# Incidence, associated risk factors, and the ideal mode of delivery following preterm labour between 24 to 28 weeks of gestation in a low resource setting

Herbert Kayiga[1]*, Diane Achanda Genevive[2], Pauline Mary Amuge[3], Josaphat Byamugisha[1], Annettee Nakimuli[1], Andrew Jones[4]

1 Department of Obstetrics and Gynecology, Makerere University College of Health Sciences, Kampala, Uganda, 2 Kawempe National Referral Hospital, Kampala, Uganda, 3 BAYLOR College of Medicine, Kampala, Uganda, 4 University of Manchester, Manchester, United Kingdom

* hkayiga@gmail.com

## Abstract

### Background

Preterm labour, between 24 to 28 weeks of gestation, remains prevalent in low resource settings. There is evidence of improved survival after 24 weeks though the ideal mode of delivery remains unclear. There are no clear management protocols to guide patient management. We sought to determine the incidence of preterm labour occurring between 24 to 28 weeks, its associated risk factors and the preferred mode of delivery in a low resource setting with the aim of streamlining patient care.

### Methods

Between February 2020 and September 2020, we prospectively followed 392 women with preterm labour between 24 to 28 weeks of gestation and their newborns from admission to discharge at Kawempe National Referral hospital in Kampala, Uganda. The primary outcome was perinatal mortality associated with the different modes of delivery. Secondary outcomes included neonatal and maternal infections, admission to the Neonatal Special Care Unit (SCU), need for neonatal resuscitation, preterm birth and maternal death. Chi-square test was used to assess the association between perinatal mortality and categorical variables such as parity, mode of delivery, employment status, age, antepartum hemorrhage, digital vaginal examination, and admission to Special Care unit. Multivariate logistic regression was used to assess the association between comparative outcomes of the different modes of delivery and maternal and neonatal risk factors.

### Results

The incidence of preterm labour among women who delivered preterm babies between 24 to 28 weeks was 68.9% 95% CI 64.2–73.4). Preterm deliveries between 24 to 28 weeks contributed 20% of the all preterm deliveries and 2.5% of the total hospital deliveries.

**Data Availability Statement:** All relevant data are within the manuscript and its S1 File.

**Funding:** "This project was supported by HEPI project under Makerere University College of Health Sciences through The Fogarty International Center of the National Institutes of Health, U.S. Department of State's Office of the U.S. Global AIDS Coordinator and Health Diplomacy (S/GAC), and President's Emergency Plan for AIDS Relief (PEPFAR) under Award Number 1R25TW011213. We obtained 4000 USD from HEPI to run this project. The content is solely the responsibility of the authors and does not necessarily represent the official views of the National Institutes of Health. None of the authors were paid to undertake this study in form of salaries".

**Competing interests:** The authors have declared that no competing interests exist. The address of Ubunifu Afrika Limited was just used by one of the authors as a contact address but we have no dealings in any regard with this company in our study. We therefore declare that Ubunifu Afrika Limited does not in any way alter our adherence to any of the PLOS ONE policies on sharing data and materials. Diane Achanda Genevieve had used the address of her husband as she was changing employment during the study period. She has never worked otherwise under Ubunifu Afrika Limited. She's currently a Nutritionist at Kawempe National Referral Hospital. Her address has been changed accordingly.

**Abbreviations:** **HDU**, High Dependency Unit; **HIV**, Human Immunodeficiency Virus; **IRB**, Institutional Review Board; **IUFD**, Intrauterine Fetal Death; **IVF**, Invitro Fertilization; **PPROM**, Preterm Premature Rupture of Membranes; **PROM**, Premature Rupture of Membranes; **SCU**, Neonatal Special Care Unit.

Preterm labour was independently associated with gravidity (p-value = 0.038), whether labour was medically induced (p-value <0.001), number of digital examinations (p-value <0.001), history of vaginal bleeding prior to onset of labour (p-value < 0.001), whether tocolytics were given (p-value < 0.001), whether an obstetric ultrasound scan was done (p-value <0.001 and number of babies carried (p-value < 0.001). At multivariate analysis; multiple pregnancy OR 15.45 (2.00–119.53), p-value < 0.001, presence of fever prior to admission OR 4.03 (95% CI .23–13.23), p-value = 0.002 and duration of drainage of liquor OR 0.16 (0.03–0.87), p-value = 0.034 were independently associated with preterm labour. The perinatal mortality rate in our study was 778 per 1000 live births. Of the 392 participants, 359 (91.5%), had vaginal delivery, 29 (7.3%) underwent Caesarean delivery and 4 (1%) had assisted vaginal delivery. Caesarean delivery was protective against perinatal mortality compared to vaginal delivery OR = 0.36, 95% CI 0.14–0.82, p-value = 0.017). The other protective factors included receiving antenatal corticosteroids OR = 0.57, 95% CI 0.33–0.98, p-value = 0.040, Doing 3–4 digital exams per day, OR = 0.41, 95% 0.18–0.91, p-value = 0.028) and hospital stay of > 7 days, p value = 0.001. Vaginal delivery was associated with maternal infections, postpartum hemorrhage, and admission to the Special Care Unit.

## Conclusion

Caesarean delivery is the preferred mode of delivery for preterm deliveries between 24 to 28 weeks of gestation especially when labour is not established in low resource settings. It is associated with lesser adverse pregnancy outcomes when compared to vaginal delivery for remote gestation ages.

## Introduction

Preterm labour, which is the spontaneous onset of frequent contractions of significant intensity leading to progressive cervical effacement and dilatation before 37 weeks, complicates 5–10% of all pregnancies globally [1]. Preterm labour leads to nearly half of all preterm deliveries [2]. The incidence of preterm labour in Sub-Saharan Africa is between 15–38% [3] as compared to Europe 5–11% [4] and 10–15% in the United States [5]. The high burden of preterm labour in Sub-Saharan Africa has also been attributed to the higher burden of intrauterine infections that are often mismanaged or present late to the clinical teams as patients first seek native remedies prior to coming to hospitals. There is also limited access to tocolytics in low resource settings [6].

Preterm births have been categorized as extreme preterm (< 28 weeks), very preterm (28-<32 weeks) and moderate preterm (32-<37 weeks [7]. The very preterms (between 28 -< 32 weeks) and extreme preterms (below 28 weeks) account for 0.5% of all births, and have the lowest survival chances [8]. In low resource settings, only ten percent survive and lifetime physical, and neurological disabilities are often present among survivors [9]. The chances of an African preterm baby dying from complications of prematurity are 12 times more than those of a European baby [3]. In Uganda, of the 1,665,000 annual births, 226,000 are born preterm of which 12,500 die from complications of prematurity [10].

In Uganda, though the exact incidence of preterm labour occurring between 24 to 28 weeks of gestation is not known, a significant proportion of preterm deliveries occur below 28 weeks [11], These are considered as inevitable pregnancy losses. Their management is usually

expectant and these preterm babies often die due to inadequate pre-delivery preparation. There is evidence that with advances in technology, corticosteroids, tocolytics and antibiotics, preterm babies can have improved survival rates after 24 weeks [12].

Though Caesarean delivery is commonly used for extreme preterm delivery in resource rich settings, the route of delivery is controversial and determined on a case-by-case basis according to the Cochrane systematic review [13]. The ideal mode of delivery in gestation age between 24 to 28 weeks remains unclear in low resource settings. With a low caesarean section rate of 7.3% reported in Africa as compared to 25% in Europe and 32.5% in North America, a significant number of mothers scheduled for Caesarean section, deliver vaginally before accessing theatre especially among the rural poor in Africa [14].

It's against this backdrop that we sought to determine the incidence, associated risk factors and ideal mode of delivery in preterm labour occurring between 24 to 28 weeks with the aim of streamlining patient care for those at risk to improve their pregnancy outcomes.

## Materials and methods

### Study design

We conducted a hospital based prospective cohort study of 392 women with preterm labour between 24 to 28 weeks of gestation between February 2020 and September 2020 at Kawempe National Referral Hospital, Kampala Uganda. Eligible participants and their babies were followed from admission to discharge from the hospital.

Though a randomized clinical trial would have minimized bias, it was rendered unethical according to the local IRB as it would subject mothers in the Caesarean arm to undue risk in the subsequent pregnancies especially when the inter-pregnancy interval was less than 18 months. It's to this end that the study design was a prospective observation cohort to answer the specific study objectives. According to the Kawempe National Referral hospital records, of 24,526 deliveries in 2019, 2,784 were preterm. This shows that preterm labour was prevalent; a prospective cohort study was therefore feasible in the local setting for gestation age between 24 and 28 weeks.

### Study setting

This study was conducted in the obstetric wards of Kawempe National Referral Hospital. Kawempe National Referral Hospital is located about 12 kilometers (7.5 miles) from Kampala City Center, by road, north of the city's central business district, along the Kampala-Gulu highway in Uganda. It's located in Kawempe Division. It is one of the teaching hospitals for Makerere University College of Health Sciences (MakCHS). Kawempe National Referral Hospital is a government-funded hospital offering mainly free maternal and newborn healthcare services, with a bed capacity of about 900, although the patient volume served by the hospital always exceeds this capacity. The hospital serves a population of approximately 4.5 million.

"The study units included the labour ward, emergency gynaecology unit, postnatal ward, labour ward operating theatre, HDU (High Dependency Unit) and the Neonatal Special Care Unit". The units operate 24 hours per day.

"In the labour ward and emergency gynaecology unit", patients with signs of preterm labour were recruited and followed up to discharge from hospital. Women who progressed to have vaginal delivery were monitored in the labour wards. Those who had planned or emergency Caesarean delivery, were followed up to the labour ward operating theatres. The operating theatres have two operating tables. The operating theatres function 24 hours with two dedicated shifts (2 Obs/Gyn residents, 2 Intern doctors, 4 scrub nurses, 2 Anesthetic officers and 1 cleaner). Patients with complications such as pre-eclampsia with signs of severity,

postpartum haemorrhage, and caesarean hysterectomy are kept in the High Dependency Unit (HDU) for closer monitoring and critical care. This unit has six beds and per shift, there is an obstetrician, one Ob/Gyn resident, and four nurses. Preterm babies with need for more than basic resuscitation were admitted in the Neonatal Special care unit. This unit admits more than 30 preterm babies from within the hospital and from outside daily. The unit is managed by two Neonatologists, three Neonatology fellows, three paediatric residents, three intern doctors, and five nurses. The hospital has also just launched a Neonatology fellowship program through Makerere University. In its first year of enrollment, it has three fellows available at least 12 hours a day.

## Participants

The criteria for identifying study participants was based on experience of any of the following conditions: a history of painful contractions of progressive intensity, lower back or abdominal pain, bloody mucus discharge, leakage of amniotic fluid between 24 to 28 weeks of gestation. The research assistants identified and approached the potential participants from the labour ward and the emergency gynaecology unit. Interested individuals were given all the required information to enable them make an informed decision to participate in the study. They were reassured that participating in the study was voluntary and that they could opt out of the study whenever they felt like without it affecting the care they were receiving from the hospital.

Upon consenting to participate in the study, a detailed patient history was obtained. This included ascertaining presence or absence of labour pains, show, leakage of amniotic fluid, evidence of cervical effacement and dilation on physical examination to confirm the diagnosis of labour. When premature rupture of membranes was suspected, a sterile speculum examination was done to confirm PROM and also to exclude cord prolapse. Data was also retrieved from the patients' records. In cases of under or over documentation in the patients' records, both the participants and the care team were consulted for clarity.

Any woman admitted with or without preterm labour with a single or multiple viable fetuses between 24 to 28 weeks of gestation who consented to participate in the study were recruited.

Women with preterm labour between 24 to 28 weeks of gestation with fetal demise (IUFD) or with congenital anomalies were excluded. Women diagnosed and managed for preterm labour in other facilities but referred after deliveries were also excluded from our study.

**Categorization of the participants.**   Participants were categorized either as having "preterm labour" using a proxy of history labour like pains, leakage of amniotic fluid or having progressive cervical dilatation and effacement. When these presentations were absent, we categorized them as "No preterm labour." This occurred in participants who ended up with medical inductions for indications like severe preeclampsia, and intrauterine growth restriction. Gestational age was determined using the participants' first day of the last normal menstrual period or by any available antenatal obstetric ultrasound scan reports. We preferred first trimester ultrasound scan reports but in their absence, we used any available reports to estimate the gestation age.

## Outcome variables

**Primary outcome.**   Perinatal mortality defined as any fresh stillbirth or neonatal death within the first seven days of life.

**Neonatal variables.**   Admission to special care unit (SCU), Preterm birth, neonatal infections (defined by evidence of high grade fever, failure to feed, vomiting or convulsions), neonatal convulsions, Apgar score at delivery (this assessed the baby's appearance, Pulse, Grimace,

Activity and Respiration. The score is given at 1 and 5 minutes out of 10, and assesses the immediate status of the newborn at birth), and need for neonatal resuscitation.

**Maternal variables.** Parity, age, socio-economic status, prior obstetric history, HIV sero-status, and mode of delivery.

**Maternal outcome variables.** Duration of hospital stay, evidence of chorioamnionitis (defined by fever and clinical exam consistent with endometritis, initiation of antibiotics), postpartum hemorrhage, wound sepsis, wound separation, and maternal death.

## Sample size calculation

**Sample size estimation for primary outcome.** Using OpenEpi sample size calculator with an estimated perinatal mortality rate of 5% among those delivered vaginally compared to 15% among those delivered by caesarean, with an estimated difference of 10%, a sample size needed was 320 women with 80% power and significance at p<. 05.

Taking into account of a 20% loss to follow up, we recruited 392 participants into the study.

## Sample size calculation for factors associated with preterm labour (secondary outcomes)

Using a study by Wagura [3], that looked at the factors associated with preterm births at Kenyatta National hospital in Kenya, women with parity> 4 were 5 times more likely to have preterm births than those <4 (OR 4.709, p = 0.019). Using OpenEpi info sample size calculator, with OR of 4.709, power of 80%, two sided confidence level of 95% and level of significance of <0.05, we got a sample size for associated factors of 180 participants. We chose the larger sample size of 392, to determine both our primary and secondary objectives for the study.

## Staff training and recruitment

Nurse-midwives familiar with the local hospital setting were enrolled for research training for three days. They were trained on how to identify potential clients. They were also trained on participant recruitment while observing the research ethics in accordance to the Declaration of Helsinki [15]. Sixteen research assistants were then selected. Eight nurse-midwives collected data during the day while the other worked at night throughout the week during the study period. The nurse-midwives were also trained on how to identify emergencies like cord prolapse, antepartum haemorrhage, impending uterine rupture and the procedures to undertake so as to inform the obstetric team on duty so that the affected participants obtained timely emergency care to optimize their pregnancy outcomes.

## Sampling procedure

All eligible mother-baby pairs who met the inclusion criteria were recruited upon giving informed consent and were thereafter followed up from "the labour and emergency gynaecology wards to the postnatal wards" up to discharge from hospital. A pretested standard questionnaire was administered to the mothers with or without preterm labour between 24 to 28 weeks. Baby and mothers' medical records were retrieved whenever additional information was needed for clarity. Trained staff using pretested data extraction tools, collected participants' socio-demographic characteristics, neonatal and maternal variables.

**Management options.** Whenever patients came in with preterm labour between 24 to 28 weeks with no signs of fetal or maternal distress, they were managed conservatively or expectantly. Tocolytic drugs, corticosteroids and antibiotics were given to prolong the pregnancy with an ultimate mode of delivery being vaginal when labour set in. However when there were

signs of fetal or maternal compromise or impending signs of infections, then active management (induction of labour with either Misoprostol or Pitocin infusion), or by Caesarean delivery were used. Caesarean delivery included participants scheduled for elective caesarean delivery without a trial of labour or emergency caesarean when the baby's or maternal conditions were in danger. The indications for Caesarean delivery in our study included, signs of acute chorioamnionitis, fetal malpresentation like breech presentation, severe oligohydramnios, prior Caesarean scars especially when more than one scar, and presence of non-reassuring fetal heart. Assisted vaginal delivery was instituted for some participants when they had delay in second stage and when some participants had maternal exhaustion in second stage.

**Data collection procedure.** Using a constructed data collection tool, information on each study participant was collected. This included: their bio-data, the management given (Antibiotics, corticosteroids, and tocolytics), Mode of delivery, maternal and neonatal outcomes. Upon consenting, history taking and physical examination were carried out on all study participants by the sixteen trained research assistants. These trained Nurse-midwives were familiar to the local setting. According to the planned mode of delivery by the obstetric team on duty, the participants were followed from admission up to discharge from the hospital. This was not random selection but decided as per the clinical presentation of the participants. The study ran over 24 hours, Monday up to Sunday every week. Data collected was crosschecked and entered on the same day of collection.

## Statistical analysis

Data entry was done using EPI-DATA 3.1 and analyzed using STATA version 14. Baseline characteristics were described using descriptive statistics presented in, means and percentages as appropriate. We determined the incidence of preterm labour by determining the percentage of women with preterm labour between 24 to 28 weeks of gestation out of the total number of women admitted with gestation ages between 24 to 28 weeks during the study period. We determined the incidence of preterm deliveries between 24 to 28 weeks by determining the proportion of women who delivered between 24 to 28 weeks out of the total deliveries in the hospital between February 2020 and September 2020 and expressed the incidence as a percentage of the total deliveries during the study period. Inferential statistics were used to establish the association between preterm labour and various risk factors using the appropriate statistical test. Chi-square test and Fischer's exact tests were used to assess association between preterm birth, perinatal mortality and categorical variables such as parity, age, presence of a febrile illness prior to admission, prior caesarean section scars, need for neonatal resuscitation, and neonatal convulsions. Logistic regression was used to assess association between preterm birth and perinatal mortality with maternal and neonatal numerical variables such as age, Apgar score, and duration of hospital stay. Variables with a p-value of 0.2 at bi-variate analysis were included in a multivariate logistic regression model to assess association with perinatal mortality and mode of delivery. Variables with p-value of $< 0.05$ were considered as significantly associated with perinatal mortality.

**Quality control.** Sixteen Research Assistants were trained prior to the data collection. Eight research assistants collected data during day and the other eight at night. A pilot study was carried out to pretest and modify the data collection tools. Completed data collection tools were checked for completeness and thereafter edited, coded and entered on same day of collection.

## Ethical consideration

IRB approvals were obtained from the Hospital Institutional Ethics review board with a registration number TASOREC/053/19-UG-REC-009, the Department of Obstetrics and

Gynecology, Makerere University, and written informed consent was obtained from all study participants. For the emancipated minors, we obtained assent from the care takers or guardians as recommended by the IRB.

## Results

### Baseline characteristics of participants

From February 2020 to September 2020, 15,346 women delivered 15,401 babies at Kawempe National Referral Hospital in Kampala, Uganda. There were 1,916 preterm deliveries during the study period giving an incidence of preterm birth of 12.4%. Preterm births between 24 to 28 weeks constituted 20% of the total number of preterms delivered during the study period. The overall incidence of preterm births between 24 to 28 weeks of gestation at Kawempe National Referral hospital was 2.5%. The incidence of preterm labour among women who delivered between 24 to 28 weeks in the study population during this period was 68.9%, "95% CI: 64.2–73.4)". Among the reasons for those who were medically induced between 24 to 28 weeks of gestation and delivered were hypertensive disorders of pregnancy (like preeclampsia), Diabetes mellitus, and intrauterine growth restriction.

We recruited 392 eligible participants consecutively. We had no loss to follow up in our study. The participants were normally distributed with mean age of 26(±5.8) years; the youngest participant being 14 years and the oldest being 42 years. The median gravidity was 2 with an interquartile range of 3.2. Most of the participants (64.0%) were either unemployed or housewives. The HIV prevalence in our study was 6.4%. About 2.3% of the mothers had previous operations on the cervix (cervical cerclage) for cervical incompetence. In our study, 85.5% of the mothers had live babies as the outcome of the previous pregnancy. However, 37% of the live babies were premature babies. Majority of the study participants 359/392 (91.5%) had vaginal delivery. Caesarean delivery accounted for 29/392 (7.3%) while Assisted vaginal delivery contributed to 4/392 (1%). Caesarean section rates were five times higher in the group that had no preterm labour compared to mothers who had preterm labour (3.7% vs. 15.6%).

Half of the mothers had history of leakage of amniotic fluid before onset of labour (PROM) and all of them had preterm birth (Table 1).

From the baseline and obstetric characteristics, the following variables were independently associated with the preterm labour: gravidity (p-value = 0.038), whether labour was medically induced (p-value <0.001), number of digital examinations (p-value <0.001), history of vaginal bleeding prior to onset of labour (p-value < 0.001), whether drugs to delay labour were given (p-value < 0.001), whether an obstetric ultrasound scan was done (p-value <0.001 and number of babies carried (p-value <0.001) (Tables 1 and 2).

### Impact of mode of delivery on perinatal mortality

There were 305/392 perinatal deaths 77.8% 95% CI 73.4–81.7). This translates into a perinatal mortality rate of 778 per 1000 live births, of which 186 (61%) were fresh stillbirths, 100 (33%) were macerated stillbirths and 18 (6%) were early neonatal deaths. Caesarean delivery, leakage of amniotic fluid for more than seven days, and administration of corticosteroids were significantly associated with perinatal mortality (Table 3).

At multivariable analysis, Caesarean delivery (compared to vaginal delivery) OR = 0.36, p-value = 0.017 was protective against perinatal mortality. The other protective factors included receiving corticosteroids OR = 0.57 p-value = 0.040, doing 3–4 digital exams per day (the reference group is one where no digital exam was done), OR = 0.41, p-value = 0.028) and hospital stay of more than 7 days, p value = 0.001 (Table 4).

**Table 1. Baseline and obstetric characteristics of 392 women admitted between 24 to 28 weeks of gestation at Kawempe National Referral Hospital, Kampala, Uganda.**

| Variable | No Preterm Labour (n = 122) n (%) | Preterm Labour (n = 270) n (%) | p-value |
|---|---|---|---|
| **Age** | | | |
| < 18 years | 3(2.5) | 14(5.2) | 0.140 |
| 18–24 years | 47(38.5) | 117(43.3) | |
| 25–35 years | 51(41.8) | 112(41.5) | |
| > 35 years | 21(4.6) | 27(10.0) | |
| **Occupation** | | | |
| Employed | 49(40.2) | 92(34.1) | 0.245 |
| Unemployed/housewife | 73(59.8) | 178(65.9) | |
| **Does the participant smoke?** | | | |
| Yes | 16(13.1) | 36(13.3) | 0.953 |
| No | 106(86.9) | 234(86.7) | |
| **Monthly Income** | | | |
| <50,000 UGX | 9(18.4) | 24(26.1) | 0.415 |
| 50,000-<500,000/ = | 34(69.4) | 60(65.1) | |
| 500,000-<1,000,000/ = | 5(10.2) | 08(8.7) | |
| >1,000,000/ = | 1(2.0) | 00(0.0) | |
| **Gravidity** | | | |
| Prime gravida | 34(27.9) | 85(31.5) | **0.038** |
| Gravida 2–3 | 39(32.0) | 111(41.1) | |
| Gravida≥4 | 49(40.2) | 74(27.4) | |
| **HIV Status** | | | |
| Positive | 7(5.7) | 18(6.7) | 0.742 |
| Negative | 115(94.3) | 252(93.3) | |
| **Number of previous uterine scars** | | | |
| None | 104(85.2) | 233(86.3) | 0.892 |
| One | 14(1.5) | 27(10.0) | |
| Two or more | 4(3.3) | 10(3.7) | |
| **History of leakage of amniotic fluid** | | | |
| Yes | 28(22.9) | 76(28.1) | 0.281 |
| No | 94(77.1) | 194(71.9) | |
| **Outcome of previous pregnancy** | | | |
| Live Premature | 32(36.4) | 55(29.7) | 0.265 |
| Live full term | 40(45.5) | 105(56.7) | |
| Fresh Stillbirth | 10(11.4) | 17(9.2) | |
| Macerated Stillbirth | 3(3.4) | 1(0.5) | |
| Early Neonatal Death | 2(2.3) | 6(3.2) | |
| Congenital anomalies | 1(1.1) | 1(0.5) | |

p-value<0.05 were significant.

## Effect of mode of delivery on secondary maternal outcomes

Vaginal delivery was associated with increased maternal morbidity. We had no maternal deaths in our study. In our study, 2.6% (n = 10) of the mothers developed complications, 40% were local infections while 60% were postpartum haemorrhage. All the cases of postpartum haemorrhage followed vaginal delivery.

**Table 2. Baseline and obstetric characteristics of 392 women admitted between 24 to 28 weeks of gestation at Kawempe National Referral Hospital, Kampala, Uganda.**

| Variable | No Preterm Labour (n = 122) n(%) | Preterm Labour (n = 270) n(%) | p-value |
|---|---|---|---|
| **History of operation on the cervix** | | | |
| No | 121(99.2) | 267(98.8) | 0.332 |
| Yes | 1(0.8) | 3(1.2) | |
| **History of fever prior to admission** | | | |
| No | 110(90.1) | 216(80.0) | 0.13 |
| Yes | 12(9.9) | 54(20.0) | |
| **Comorbidity at admission** | | | |
| Malaria | 5(41.7) | 18(33.3) | 0.856 |
| Urinary tract infections | 5(41.7) | 25(46.3) | |
| Others | 2(16.6) | 11(20.4) | |
| **History of vaginal bleeding prior to labour onset** | | | |
| No | 98(80.3) | 168(62.2) | <0.001 |
| Yes | 24(19.7) | 102(27.8) | |
| **Labour medically induced** | | | |
| No | 70(57.4) | 257(95.2) | <0.001 |
| Yes | 52(46.6) | 13(4.8) | |
| **History of leakage of Amniotic fluid** | | | |
| Present | 28(22.9) | 76(28.1) | 0.281 |
| Absent | 94(77.1) | 194(71.9) | |
| **Duration of leakage of Amniotic fluid** | | | |
| <24 hours | 17(35.4) | 81(54.7) | 0.053 |
| 24<48 hours | 21(43.8) | 54(36.5) | |
| 2–7 days | 6(12.5) | 8(5.4) | |
| >7days | 4(8.3) | 5(3.4) | |
| **Color of Amniotic fluid** | | | |
| Clear | 43(89.6) | 126(85.1) | 0.605 |
| Light meconium | 5(10.4) | 20(13.5) | |
| Thick meconium | 0(0.0) | 2(1.3) | |
| **Smell of Amniotic fluid** | | | |
| Offensive | 7(14.6) | 15(10.1) | 0.396 |
| Not offensive | 41(85.4) | 133(89.9) | |
| **Speculum exam done** | | | |
| Yes | 22(18.0) | 55(25.6) | 0.590 |
| No | 100(82.0) | 215(74.4) | |
| **Digital exams per day** | | | |
| None | 71(58.2) | 97(35.9) | 0.001 |
| 1–2 | 42(34.4) | 144(53.3) | |
| 3–4 | 9(7.4) | 27(10.0) | |
| ≥ 5 | 0(0.0) | 2(0.8) | |
| **Fetal Heart Present** | | | |
| Yes | 3(2.5) | 12(4.4) | 0.343 |
| No | 119(97.5) | 258(95.6) | |
| **Obstetric Ultrasound scan was done** | | | |
| No | 13(10.7) | 115(42.6) | <0.001 |
| Yes | 109(89.3) | 155(57.4) | |
| **Number of babies** | | | |
| Singleton (one) | 114(93.1) | 204(75.6) | <0.001 |
| More than one baby | 7(5.9) | 48(17.8) | |
| Not sure | 1(0.8) | 18(6.7) | |
| **Drugs given to delay labour** | | | |
| No | 54(44.3) | 222(82.2) | <0.001 |
| Yes | 68(55.7) | 48(17.8) | |

p-value<0.05 were significant.

**Table 3. Comparison of obstetric characteristics and perinatal mortality among 392 women who had preterm delivery between 24 to 28 weeks of gestation at Kawempe National Referral Hospital, Kampala, Uganda.**

| Variable | Baby Alive (n = 87) n(%) | Baby Dead (n = 305) n(%) | OR (95% CI) | p-value |
|---|---|---|---|---|
| **Mode of delivery** | | | | |
| Spontaneous Vaginal Delivery | 74(85.1) | 285(93.4) | 1.0 | - |
| Assisted Vaginal Delivery | 1(1.1) | 3(1.0) | 0.78(0.08–7.60) | 0.830 |
| Caesarean | 12(13.8) | 17(5.6) | 0.36(0.17–0.80) | **0.012** |
| **History of leakage of Amniotic fluid** | | | | |
| Yes | 19(21.8) | 85(27.9) | 1.0 | 0.261 |
| No | 68(78.2) | 220(72.1) | 0.81(0.50–1.31) | |
| **Duration of leakage of Amniotic fluid** | | | | |
| <24 hours | 19(40.3) | 79(53.0) | 1.0 | - |
| 24<48hours | 18(38.3) | 57(38.3) | 0.76(0.37–1.58) | 0.464 |
| 2–7 days | 5(10.6) | 9(6.0) | 0.43(0.13–1.44) | 0.172 |
| >7days | 5(10.6) | 4(2.7) | 0.92(0.05–0.78) | **0.022** |
| **Color of Amniotic fluid** | | | | |
| Clear | 40(85.1) | 129(86.6) | 1.0 | - |
| Light meconium | 7(14.9) | 18(12.1) | 0.78(0.31–2.04) | 0.638 |
| Thick meconium | 0(0.0) | 2(1.3) | 1.0 | - |
| **Smell of Amniotic fluid** | | | | |
| Offensive | 6(12.8) | 16(10.3) | 1.0 | - |
| Not offensive | 41(87.2) | 133(89.7) | 1.22 (0.45–3.31) | 0.701 |
| **Fever prior to admission** | | | | |
| No | 69(79.3) | 257(84.3) | 1.0 | |
| Yes | 18(20.7) | 48(15.7) | 0.71(0.39–1.31) | 0.276 |
| **Comorbidity at admission** | | | | |
| Malaria | 7(38.9) | 16(33.3) | 1.0 | - |
| Urinary tract infections | 8(44.4) | 22(45.8) | 1.20(0.36–4.00) | 0.763 |
| Others | 3(16.7) | 10(20.8) | 1.45(0.30–6.98) | 0.647 |
| **History of vaginal bleeding prior to onset of labour** | | | | |
| No | 62(71.2) | 204(66.9) | 1.0 | - |
| Yes | 25(28.8) | 101(33.1) | 1.23(0.73–2.07) | 0.440 |
| **Labour medically induced** | | | | |
| No | 77(88.5) | 250(82.0) | 1.0 | |
| Yes | 10(11.5) | 55(18.0) | 1.69(0.82–3.48) | 0.148 |
| **Speculum exam done** | | | | |
| Yes | 16(18.4) | 61(20.0) | 1.0 | |
| No | 71(81.6) | 244(80.0) | 0.90(0.49–1.66) | 0.739 |
| **Digital exams per day** | | | | |
| None | 35(40.3) | 133(43.6) | 1.0 | |
| 1–2 | 38(43.7) | 148(48.5) | 1.02(0.61–1.71) | 0.925 |
| 3–4 | 13(14.9) | 23(7.5) | 0.46(0.21–1.01) | 0.053 |
| ≥5 | 1(1.1) | 1(0.4) | 0.26(0.02–4.31) | 0.349 |
| **Were corticosteroids given?** | | | | |
| No | 35(40.2) | 154(50.5) | 1.0 | |
| Yes | 54(59.8) | 151(49.5) | 0.60(0.37–0.97) | **0.039** |
| **Were antibiotics given?** | | | | |
| No | 37(42.5) | 154(50.5) | 1.0 | |
| Yes | 50(57.5) | 151(49.5) | 0.72(0.45–1.17) | 0.190 |

*(Continued)*

**Table 3.** (Continued)

| Variable | Baby Alive (n = 87) n(%) | Baby Dead (n = 305) n(%) | OR (95% CI) | p-value |
|---|---|---|---|---|
| **Tocolytics given** | | | | |
| No | 62(71.3) | 214(70.2) | 1.0 | - |
| Yes | 25(28.7) | 91(29.8) | 1.05(0.62–1.78) | 0.843 |
| **Number of babies** | | | | |
| Singleton (one) | 69(79.3) | 249(81.6) | 1.0 | - |
| >1 baby | 14(16.1) | 41(13.4) | 0.81(0.42–1.57) | 0.537 |
| Not sure | 4(4.6) | 15(4.9) | 1.04(0.33–3.23) | 0.947 |

p-value<0.05 statistically significant.

**Impact of mode of delivery on admission to Special Care Unit (SCU).** Most of the babies (n = 231, representing 58.9%) required SCU admission, with 31% of them spending more than 7 days. Prematurity was the commonest reason for admission 193/231 (93.1%). The other indications were Asphyxia (6.1%) and Respiratory Distress Syndrome (0.9%). Vaginal delivery accounted 89.1% of the admissions to SCU followed by Caesarean delivery (9.5%). Babies who were admitted to SCU longer than 7 days had more chances of surviving thereafter as compared to those admitted < 24 hours p-value <0.001.

**Impact of corticosteroid administration on perinatal mortality.** Though all participants were eligible for corticosteroid use, 205/392 (52.2%) received in our study. Corticosteroid use significantly improved the perinatal outcomes following preterm labour. Preterm babies who received corticosteroids intrauterine had 40% increased chance of surviving as compared to those who didn't receive the corticosteroids(p-value = 0.039) (Table 3).

## Predictors of preterm delivery

At bivariate analysis, Age (p-value = 0.140), gravidity (p-value = 0.038), history of a fever prior to admission (p-value = 0.13), history of vaginal bleeding prior to onset of labour (p-value <0.001), whether labour was medically induced (p-value <0.001), number of digital examination in a day (p-value = 0.001), whether an obstetric ultrasound was done (p-value < 0.001), number of babies carried (p-value <0.001) and whether drugs to delay labour were administered (p-value <0.001) were < 0.2 (Tables 1 and 2) and were further evaluated at multivariate analysis. More than one baby carried, presence of fever prior to admission and duration of leakage of amniotic fluid were independently associated with preterm labour (Table 5).

**Table 4. Multivariate association between perinatal mortality and independent variables for mothers delivering between 24 to 28 weeks of gestation at Kawempe National Referral Hospital, Uganda.**

| Variable | OR (95% CI) | p-value |
|---|---|---|
| Caesarean delivery | 0.36, 95% CI 0.14–0.82, | **0.017** |
| Spontaneous vaginal delivery | 1 | |
| Receiving corticosteroids | 0.57, 95% CI 0.33–0.98 | **0.040** |
| Did not receive corticosteroids | 1 | |
| Doing 3–4 digital exams per day | 0.41, 95% 0.18–0.91 | **0.028** |
| No digital exam done | 1 | |
| Hospital stay > 7 days | 0.46, 95% 0.15–0.82 | **0.001** |
| Hospital stay < 1 day | 1 | |

## Multivariate analysis

**Table 5. Multivariate association between preterm labour and independent variables for mothers delivering between 24 to 28 weeks of gestation at Kawempe National Referral Hospital, Uganda.**

| Variable | OR (95% CI) | p-value |
|---|---|---|
| More than one baby | 15.45(2.00–119.53) | <0.001 |
| Singleton delivery | 1 | |
| Presence of fever prior to admission | 4.03(0.23–13.23) | 0.002 |
| No fever prior to admission | 1 | |
| Leakage of amniotic fluid | 0.16(0.03–0.87) | 0.034 |
| No leakage of amniotic fluid | 1 | |

## Discussion

Evidence shows that early recognition of preterm labour optimizes fetal outcomes by allowing for timely transfer of women with preterm labour to facilities with advanced neonatal care services, institution of magnesium sulphate for neuro-protection and corticosteroids for lung maturity [16]. With the uncertainty surrounding the choice of delivery method, even in the developed countries [17], this study sought to determine the incidence of preterm labour, its associated risk factors and the ideal mode of delivery for deliveries before 28 weeks of gestation in a low resource setting.

In our study, the incidence of preterm labour among women who delivered between 24 to 28 weeks of gestation was 68.9%. Preterm delivery between 24 to 28 weeks of gestation contributed 20% of all preterm deliveries during the study period. The overall incidence of preterm deliveries between 24 to 28 weeks at Kawempe National Referral Hospital was 2.5%. More than one baby carried, presence of fever prior to admission and duration of leakage of amniotic fluid were independently associated with preterm labour. In our study, 77.8% of the babies died translating into a perinatal mortality rate of 778 per 1000 live births, of which 61% were fresh stillbirths. The mode of delivery had a statistically significant impact on perinatal mortality. Caesarean delivery was protective against perinatal mortality in these preterm babies. Other protective factors included administration of corticosteroids, conducting three or more digital examinations per day to monitor labour progress and when babies were admitted to the Special Care Unit for more than a week. Vaginal delivery was associated with more adverse pregnancy outcomes compared to Caesarean delivery. Nearly 90% of the admission to the Special Care Unit followed vaginal delivery.

The incidence of preterm birth between 24 to 28 weeks of 2.5% is higher than that of 1.1% in China [18], and 2% reported in the United Kingdom [19] and 2% in the United States [19, 20]. The higher incidence could have followed the high turnover of deliveries at Kawempe National Referral hospital of over 30,000 deliveries per year. The higher incidence could also have followed the fact that in our study, gestation age was mainly determined by using the first day of the last menstrual period. Obstetric ultrasound scans are paid for by the patients. In the event that the patients had wrong dates, we could have over recruited participants into the study. The increasing incidence of the preterm labour could also have been as a result of increasing maternal age at first birth. In our study, the oldest participant had 42 years in her first pregnancy. There is also a notable growth in reproductive medicine in Uganda with now over ten Invitro Fertilization (IVF) centres offering these services. There have been also increasing survival rates especially after 26 weeks with the

advancement in the neonatal care locally. Currently a neonatology fellowship program runs at Kawempe National Referral Hospital.

In our study, multiple pregnancy, having antepartum haemorrhage, gravidity, prior history of leakage of amniotic fluid and presence of a febrile illness were the commonest risk factors for preterm labour between 24 to 28 weeks of gestation. About half of all the patients who presented with preterm labour had preterm premature rupture of membranes. All of these participants ended up with preterm delivery. Shariati et al, [21] also reported similar findings, where 40–50% of the women who presented with preterm labour between 24 to 33 weeks of gestation in Iran had premature rupture of membranes. Robinson [22], also reported that multifetal pregnancy, preterm premature rupture of membranes and infections were risk factors too that were associated with preterm labour. Bouchra [23], in a study conducted in Netherlands, also reported that parity/gravidity was independently associated with preterm labour at early gestation ages especially among the nulliparous and women carrying their fifth pregnancies. Wagura [3], just like in our study, reported in his study at Kenyatta National hospital that having antepartum haemorrhage was a risk factor for preterm labour.

Though prior studies have reported that the following; unemployment [22], prior preterm delivery [24], smoking [25, 26], Age [3, 27] and maternal infections like urinary tract infections [28] were significant risk factors for preterm labour, in our study these factors were not significantly associated with preterm labour between 24 to 28 weeks of gestation.

Despite the results of improved survival rates between 24 to 28 weeks ranging from 33.3–76.3% in South Africa [29], from 0–87.4% in China [18] and from 6–94% United States [30], and from 39–93% United Kingdom [31], the survival rate at discharge at Kawempe National referral hospital was 22.2%. The high perinatal mortality could have followed the fact that the hospital doesn't have a fully equipped neonatal intensive care unit. There are no neonatal ventilators yet with limited human resource that is often overwhelmed by the high turnover of patients. In our study, 61% of the perinatal deaths were fresh stillbirths. This shows that there are significant gaps in the health system in Uganda with most of the perinatal deaths occurring Intrapartum or immediately after delivery [10]. The referral system in Uganda, is so wanting leading to many delays in the accessing the required specialized care to optimize pregnancy outcomes in these remote gestation ages. There's also minimal access to surfactant at the hospital. Since preterm births below 28 weeks are referred to as inevitable pregnancy losses, corticosteroids for lung maturation and magnesium sulphate for neuro-protection are rarely administered. This could explain the perinatal mortality of 77.5% that is higher than that reported globally of 75% [32].

In our study, 91.5% of the deliveries between 24 to 28 weeks of gestation were vaginal as compared to 7.4%. This could be so because despite the intention of the care team to have caesarean delivery for a significant number of the women with preterm labour, some delivered vaginally while on expectant management. The commonest indications for caesarean section locally between 24 to 28 weeks included prior Caesarean scars especially when a woman had more than one scar, short inter-pregnancy internal after a Caesarean delivery, breech presentation and severe oligohydramnios. Due to long queues of patients awaiting Caesarean delivery, a significant proportion of women delivered vaginally before accessing theatre space. Caesarean deliveries were more of elective than emergent in our study. The Caesarean section rate of 7.4% in our study is lower than the 51.7% reported Riskin [33]. Also in view of the recommendations of vaginal over Caesarean delivery in gestation ages remote from term [13, 34], the care team could have opted for vaginal over Caesarean delivery for gestation ages between 24 to 28 weeks.

Unlike in prior studies that showed no benefits of Caesarean over vaginal delivery [13, 17, 33, 35], Caesarean delivery was protective against perinatal mortality. Babies who were

delivered by Caesarean had a 64% chance of surviving when compared to those delivered by vaginal route (p value = 0.017). Jonas et al [36] reported an adjusted odds to perinatal mortality after Caesarean delivery of 0.55 in extreme preterms in the United States. Riskin [33] also reported that the perinatal mortality rates prior to discharge were lower after Caesarean as compared to vaginal delivery 13.2%*vs* 21.8% though it had no impact on the survival rates thereafter. Pradip and Malloy [37, 38], also reported survival advantages of Caesarean over vaginal delivery especially among babies with breech/malpresentation. Our findings could have been because Caesarean delivery was commoner among those who had preterm delivery without labour like among those with severe oligohydramnios, severe preeclampsia and in presence of prior caesarean scars. Other protective factors against perinatal deaths in our study were administration of corticosteroids, admission in the Neonatal Special Care Unit for over seven days and having 3–4 digital examinations per day.

Antenatal corticosteroid administration has been reported to improve survival rates in a number of studies [18, 33, 39–41] in extreme preterms.

Whereas digital examinations have been reported to increase the risk of chorioamnionitis [39], in our study conducting 3–4 digital examinations per day was associated with increased survival rates. This could have followed the fact that in so doing cord prolapses were identified earlier and timely interventions undertaken. Impending vaginal delivery could have been identified and better prepared for than noticing when delivery of the preterm is complete with expectant management for fear of introducing infections.

Though Thomas [42] reported that Caesarean delivery in gestation age remote from 28 weeks increased maternal morbidity, yet with no added benefits when compared to vaginal delivery, in our study, vaginal delivery was associated with more adverse pregnancy outcomes than with Caesarean delivery. All the postpartum hemorrhage that occurred in our study followed vaginal delivery. Nearly 90% of all admissions to Special Care Unit followed vaginal delivery. Most of Caesarean deliveries were elective. The findings could have been different if Caesarean delivery were for emergency cases.

The strength of this study is that we were able to follow up all of our participants from admission to discharge from hospital. We were able to recruit 392 participants, making our study one of the few studies conducted in low resource settings studying a larger sample size of women with preterm labour between 24 to 28 weeks of gestation. Prior studies including those in Africa [29], Europe [17] have had participants less than 100. Whereas prior studies have focused either on risk factors [16, 18, 21, 30, 43] or mode of delivery [17, 20] in preterm labour before 28 weeks, this study aimed at both aspects so as to streamline patient care in a low resource setting.

We wish to acknowledge that obtaining comparable cohorts for both women pre-selected for Caesarean or vaginal delivery that ended up with the same planned mode of delivery was challenging. Some of the women with a vaginal delivery plan with preterm labour ended up having Caesarean deliveries for indications like non-reassuring fetal hearts, severe oligohydramnios and chorioamnionitis. In the Caesarean cohort, some of the women delivered vaginally as they waited in the long queries before accessing the operating rooms. This could have introduced selection bias into the study, placing limits on comparability of delivery modes.

We were only able to follow up the participants up to discharge from hospital. We acknowledge this as a limitation as we were unable to determine the deaths or morbidities that occur after the participants' discharge from hospital. We also acknowledge that we were unable to evaluate the babies for major morbidities like Retinopathy of prematurity, cerebral palsy, neonatal sepsis or neuro-developmental abnormalities in this current study. As a potential future research project, we shall seek ethical approval from the participants and IRBs to use the

contacts of the participants and conduct a follow up study to determine the fetal outcomes and survival rates at one year to answer this aspect.

We also acknowledge that we relied majorly on participant's history and clinical examination to confirm the diagnosis of preterm labour and to determine the gestation age. This could have led to over or under inclusion of relevant cases into the study. Obtaining early obstetric scans is still very challenging in our setting as patients have to pay for the services.

## Conclusion

Preterm labour between 24 to 28 weeks remains prevalent in our setting contributing 2.5% of all deliveries. Preterm labour was independently associated with multiple pregnancy, leakage of amniotic fluid and presence of a fever prior to admission. Caesarean delivery was protective against perinatal mortality in preterm delivery between 24 to 28 weeks of gestation. Antenatal Corticosteroid administration and carrying 3–4 digital examinations per day were protective against perinatal mortality. Vaginal delivery for remote gestation ages was associated with more adverse maternal and fetal outcomes than with Caesarean delivery. In low resource settings, Caesarean delivery is the safer mode of delivery for preterm births between 24 to 28 weeks.

## Supporting information

**S1 File.**
(DOCX)

## Acknowledgments

We would like to thank the research team that made this research project a reality. Special appreciation goes to the research assistants who collected the data tirelessly. The research team included; Madinah Namufumba, Nakizibu Sumayah, Birungi Harriet, Tushabe Lillian, Nakidde Edith, Mpoyenda Lairatu, Nalubula Harriet, Aheebwa Lilian, Khalayi Zainah, Taaka Beatrice, Birungi Barabara, Akello Florence, Aanyu Martha, Akatuhwera Anna, Namusisi Zamu, Kabugho Amina, and Nabirye Teddy.

With great pleasure we appreciated the amazing work done by Dr. Andrew Edielu for developing the study database and also for the data analysis. A special vote of thanks goes to the Department of Obstetrics and Gynecology at Kawempe National Referral hospital for all the support it gave us during the study period. I'm forever grateful for the mentorship and guidance given to us from Prof Sarah Kiguli during the study. We are grateful to the administrators of the HEPI project at Makerere University College of Health Sciences for the all the guidance they have given us to make this study a success.

## Author Contributions

**Conceptualization:** Herbert Kayiga, Pauline Mary Amuge, Josaphat Byamugisha, Annettee Nakimuli.

**Data curation:** Diane Achanda Genevive.

**Formal analysis:** Herbert Kayiga.

**Methodology:** Herbert Kayiga, Pauline Mary Amuge, Josaphat Byamugisha, Andrew Jones.

**Project administration:** Diane Achanda Genevive.

**Resources:** Diane Achanda Genevive, Pauline Mary Amuge, Annettee Nakimuli.

**Supervision:** Diane Achanda Genevive, Josaphat Byamugisha, Annettee Nakimuli, Andrew Jones.

**Writing – original draft:** Herbert Kayiga.

**Writing – review & editing:** Herbert Kayiga, Diane Achanda Genevive, Pauline Mary Amuge, Josaphat Byamugisha, Annettee Nakimuli, Andrew Jones.

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
