## [Decision Letter · Decision Letter 0]

20 Apr 2021

PONE-D-21-00658

Incidence, Associated Risk factors, and The Ideal Mode of delivery following Preterm labour between 24 to 28 weeks of gestation in a Low resource setting.

PLOS ONE

Dear Dr. Kayiga,

Thank you for submitting your manuscript to PLOS ONE. After careful consideration, we feel that it has merit but does not fully meet PLOS ONE’s publication criteria as it currently stands. Therefore, we invite you to submit a revised version of the manuscript that addresses the points raised during the review process.

We look forward to receiving your revised manuscript.

Kind regards,

Antonio Simone Laganà, M.D., Ph.D.

Academic Editor

PLOS ONE

Additional Editor Comments:

The topic of the manuscript is interesting. Nevertheless, the reviewers raised several concerns: considering this point, I invite authors to perform the required major revisions.

Journal Requirements:

3. For more information on PLOS ONE's expectations for statistical reporting, please see https://journals.plos.org/plosone/s/submission-guidelines.#loc-statistical-reporting. Please update your Methods and Results sections accordingly.

"No, The funders had no role in study design, data collection and analysis, decision to publish, or preparation of the manuscript."

We note that one or more of the authors are employed by a commercial company: UbunifuAfrika Limited.

5.1. Please provide an amended Funding Statement declaring this commercial affiliation, as well as a statement regarding the Role of Funders in your study. If the funding organization did not play a role in the study design, data collection and analysis, decision to publish, or preparation of the manuscript and only provided financial support in the form of authors' salaries and/or research materials, please review your statements relating to the author contributions, and ensure you have specifically and accurately indicated the role(s) that these authors had in your study. You can update author roles in the Author Contributions section of the online submission form.

5.2. Please also provide an updated Competing Interests Statement declaring this commercial affiliation along with any other relevant declarations relating to employment, consultancy, patents, products in development, or marketed products, etc.  

6. Please amend your list of authors on the manuscript to ensure that each author is linked to an affiliation. Authors’ affiliations should reflect the institution where the work was done (if authors moved subsequently, you can also list the new affiliation stating “current affiliation:….” as necessary).

Reviewers' comments:

Reviewer's Responses to Questions

**Comments to the Author**

1. Is the manuscript technically sound, and do the data support the conclusions?

Reviewer #1: Partly

2. Has the statistical analysis been performed appropriately and rigorously? 

Reviewer #1: Yes

3. Have the authors made all data underlying the findings in their manuscript fully available?

Reviewer #1: Yes

4. Is the manuscript presented in an intelligible fashion and written in standard English?

Reviewer #1: No

5. Review Comments to the Author

Reviewer #1: Thank you for the opportunity to review the manuscript!

While this manuscript addresses an important health issue in a low resource setting, it needs a lot of work to clarify some ambiguities and be brought to a standard format for publication. Please see below my comments.

INTRODUCTION:

- Para 1, line 93please add the rate of pre-term labour in the United States.

- Para 3, line 109: the authors have indicated that preterm deliveries that occur below 28 weeks are considered as inevitable abortion. According to the universally accepted definition of abortion, an abortion is a procedure in which they use medicine or surgery to r to end a pregnancy prior to 20 weeks' gestation or when the fetus born weighs less than 500 g. Please amend your sentence. You may write “These are considered as inevitable pregnancy loss.”

- Line 114: please replace “case-to-case” with “case-by-case”.

MATERIALS AND METHODS:

- My general comment on this section is that there are several headings and subheadings that can be combined, or moved around to ensure a logical flow to the manuscript. For instance, “Study design” and “study design justification” need to be combined under one single heading. “Inclusion criteria”, “exclusion criteria” both can go under the “Participants”. My suggestion for the order of headings is as follows: Study design (containing design and its justification) - Study setting – Participants (containing inclusion and exclusion criteria, categorization of the participants) - Variables – Measures (tools) – Sample size calculation - Procedure (containing staff training, sampling procedure, management portions, data collection procedures) – Statistical analysis – Ethics. Of course, when combining headings/subheadings, repetitive sentences and information need to be removed and the information has to be presented in a logical order.

- Line 135: the study design is “prospective observation cohort”. Please amend.

- Study design, Line 15o: please amend as “The study units included the labour ward, emergency obstetrics unit,….”. Same with line 153: “in the labour ward and emergency obstetrics unit,…”

- Lines 171 and 178, please replace “gust of liquor” and drainage of liquor” with “leakage of amniotic fluid”. Please amend throughout.

- It is unclear whether the PROM was an inclusion or exclusion criteria. In “participants” section it is mentioned that “The criteria for identifying study participants was based on experience of any of the following conditions: a history of painful contractions of progressive intensity, lower back or abdominal pain, bloody mucus discharge, gush of liquor between 24 to 28 weeks of gestation.”. However, in the next page, line 181, it is indicated that “When premature rupture of membranes was suspected, a sterile speculum examination was done to exclude PROM or cord prolapse.” Please clarify.

- Sample size calculation: The original sample size was calculated to be 320 women. Taking into account of a 20% loss to follow-up (n=64), the final sample size should be 384. However, you recruited 392 women. Please clarify.

- Quality control section: lines 297-300, please remove these sentences: “Data was backed up. The database was password protected and the participants’ records were kept under limited access in a lockable cabin. The research materials were kept under restricted access by only authorized staff for patient confidentiality and privacy.”

- Ethical consideration section: lines 305-to the end of paragraph: Please remove “Benefits and risks involved in the study were communicated to the participants. ……. not identified by names in the final report.”

RESULTS:

- Throughout the results section, the authors have reported the significance of some variables that are not actually statistically significant according to the p-value shown in the respective tables. For example, “…duration of drainage of liquor (p- value=0.05)…”.Please revise this section and amend your test report.

- Also, there are many headings and sub-headings in the “Results” section, with no particular order that make it difficult to follow the concept. Also, there is no need to explain non-significant results. For example, the heading on “Effect of Antibiotic administration on perinatal mortality” is redundant and can be removed. I suggest to combine headings related to each other and explain only significant findings in the text. For example, you may combine “Impact of mode of delivery on perinatal mortality” and “Impact of Maternal HIV status on pregnancy outcomes”.

- Line 319: please put the “95% CI 64.2-73.4” in parenthesis.

- Line 317-320: this paragraph is a bit confusing. My understanding is that during the study period, from February 2020 to September 2020, 15,346 women delivered out of which 1,916 were preterm deliveries between 24 to 28 weeks. You reported that “The incidence of preterm labour between 24 to 28 weeks in the study population during this period was 68.9%, 95% CI 64.2 – 73.4”. The incidence is very high which does not make sense. Can you please explain how you estimated the incidence? The prevalence is 12.5%, but you have indicated it was 20%! Then, the last sentence of this paragraph mentions that “The overall incidence of preterm deliveries occurring between 24 to 28 weeks of gestation during the study period at Kawempe National Referral Hospital was 2.5%.”. Please clarify and rewrite this paragraph.

- Line 325: to show mean and standard deviation, you may write it as 26(±5.8).

- Line 326-327 “The median gravidity was 2 with a variance of 3.2”. T report median, you need median and interquartile range (IQR), not a variance. Please amend.

- Section “Impact of mode of delivery on perinatal mortality”, line 353: Please remove “95% CI 73.4 – 81.7)” from the text. The same section, the result of multivariate analysis would be easier to understand and interpreted by the readers if they are presented in a table indicating the reference item for each variable.

TABLES:

There are several issues with tables that need to be fixed:

- In all tables there are non-significant p-values that have been marked by an astrix (*) as significant. On the other hand, some of the significant p-values have not been marked! Please remove the * from the non-significant p-values: table 1a, 0.14*, table 1b, 0.13*, 0.053*.

- Table 1a: variable “number of previous scare”, does this refer to scar on cervix or uterus? Please clarify. Also variable “Outcome of previous pregnancy”, please write all abbreviated terms in full (FSB, MSB, ENND). You may add the full words as footnote to the table.

- Table 1b: variables “Labour medically induced”, “Fetal Heart Present”, “Obstetric US was done” and “Drugs given to delay labor?”, please calculate percentage for each number and add them to the table. Also, please write the ‘ultrasound (US)’ in full. Remove question mark from “Drugs given to delay labor?”

- Table 2: please write the followings in full: SVD, UTI. Also, please indicate whether the “History of vaginal bleeding” was during pregnancy or in labour.

- Table 4: please add the reference group to each variable.

-

6. PLOS authors have the option to publish the peer review history of their article (what does this mean?). If published, this will include your full peer review and any attached files.

Reviewer #1: **Yes: **A/Prof Marjan Khajehei

---

## [Author Response · Author response to Decision Letter 0]

24 Jun 2021

AUTHORS’ RESPONSE TO REVIEWS

TITLE: Incidence, Associated Risk factors, and The Ideal Mode of delivery following Preterm labour between 24 to 28 weeks of gestation in a Low resource setting

Authors

Herbert Kayiga (hkayiga@gmail.com)

Diane Achanda Genevive (achandadiane@gmail.com)

Pauline Mary Amuge (paulacallista@gmail.com)

Josaphat Byamugisha (jbyamugisha@gmail.com)

Annettee Nakimuli (Annettee.nakimuli@gmail.com)

Andrew Jones (andrew.jones@manchester.ac.uk)

Version: 1

Date: 2nd June 2021

Authors’ response to reviews: See over

 2nd June 2021

TO: THE PLOS ONE EDITORIAL TEAM

Object: PONE-D-21-00658: Incidence, Associated Risk factors, and The Ideal Mode of delivery following Preterm labour between 24 to 28 weeks of gestation in a Low resource setting

With great pleasure we are thankful for your consideration of our manuscript for publication in your reputable journal. In response to the editors’ comments sent to us on 20th April 2021, we have revised the above manuscript accordingly. 

Additional Editor Comments:

The topic of the manuscript is interesting. Nevertheless, the reviewers raised several concerns: considering this point, I invite authors to perform the required major revisions. 

Journal Requirements:

 Response: The manuscript has been edited as advised.

 Response: Clarity has been added in the revised manuscript to accommodate what was done for the emancipated minors in our study. This now appears in the updated manuscript in line 302-307 as “IRB approvals were obtained from the Hospital Institutional Ethics review board with a registration number TASOREC/053/19-UG-REC-009, the Department of Obstetrics and Gynaecology, Makerere University, and written informed consent were obtained from all study participants. For the emancipated minors, we obtained assent from the care takers or guardians as recommended by the IRB”. 

3. For more information on PLOS ONE's expectations for statistical reporting, please see https://journals.plos.org/plosone/s/submission-guidelines.#loc-statistical-reporting. Please update your Methods and Results sections accordingly.

 Response: The Methods and Results sections have been updated as recommended.

"No, The funders had no role in study design, data collection and analysis, decision to publish, or preparation of the manuscript."

a. Please clarify the sources of funding (financial or material support) for your study. List the grants or organizations that supported your study, including funding received from your institution.

d. If you did not receive any funding for this study, please state: “The authors received no specific funding for this work.”

 Response: The financial disclosure statement has been amended as recommended. It now appears in line 589 to 596 of the revised manuscript as “This project was supported by HEPI project under Makerere University College of Health Sciences through The Fogarty International Center of the National Institutes of Health, U.S. Department of State’s Office of the U.S. Global AIDS Coordinator and Health Diplomacy (S/GAC), and President’s Emergency Plan for AIDS Relief (PEPFAR) under Award Number 1R25TW011213. We obtained 4000 USD from HEPI to run this project. The content is solely the responsibility of the authors and does not necessarily represent the official views of the National Institutes of Health. None of the authors were paid to undertake this study in form of salaries”.

"The authors have declared that no competing interests exist." We note that one or more of the authors are employed by a commercial company: Ubunifu Afrika Limited.

 Response: Thanks for this concern. Diane Achanda Genevieve had used the address of her husband as she was changing employment during the study period. She has never worked otherwise under Ubunifu Afrika Limited. She’s currently a Nutritionist at Kawempe National Referral Hospital. Her address has been changed accordingly. 

5.1. Please provide an amended Funding Statement declaring this commercial affiliation, as well as a statement regarding the Role of Funders in your study. If the funding organization did not play a role in the study design, data collection and analysis, decision to publish, or preparation of the manuscript and only provided financial support in the form of authors' salaries and/or research materials, please review your statements relating to the author contributions, and ensure you have specifically and accurately indicated the role(s) that these authors had in your study. You can update author roles in the Author Contributions section of the online submission form.

Response: Thanks for the advice. The funding organization had no role in the study design, data collection and analysis, decision to publish, or preparation of the manuscript and only provided financial support in the form of research materials. This has been updated in the Author contribution section. 

 Response: The funders didn’t pay the salaries for any of the authors and this is indicated in the financial disclosure section. They did not have any additional role in the study design, data collection and analysis, decision to publish, or preparation of the manuscript as well. 

5.2. Please also provide an updated Competing Interests Statement declaring this commercial affiliation along with any other relevant declarations relating to employment, consultancy, patents, products in development, or marketed products, etc. 

 Response: The address of Ubunifu Afrika Limited was just used by one of the authors as a contact address but we have no dealings of any regard with this company in our study.

 Response: The address of Ubunifu Afrika Limited was just used by one of the authors as a contact address but we have no dealings in any regard with this company in our study. We therefore declare that Ubunifu Afrika Limited does not in any way alter our adherence to any of the PLOS ONE policies on sharing data and materials.

 Response: We declare that we have no competing interests of any sort whether financial or non-financial.

6. Please amend your list of authors on the manuscript to ensure that each author is linked to an affiliation. Authors’ affiliations should reflect the institution where the work was done (if authors moved subsequently, you can also list the new affiliation stating “current affiliation:….” as necessary).

Response: The address of Diane Achanda Genevieve has been changed as shown in line 17 

Reviewers' comments:

Reviewer's Responses to Questions

Comments to the Author

1. Is the manuscript technically sound, and do the data support the conclusions?

Reviewer #1: Partly

2. Has the statistical analysis been performed appropriately and rigorously?

Reviewer #1: Yes

3. Have the authors made all data underlying the findings in their manuscript fully available?

Reviewer #1: Yes

4. Is the manuscript presented in an intelligible fashion and written in standard English?

Reviewer #1: No

5. Review Comments to the Author

Reviewer #1: Thank you for the opportunity to review the manuscript!

While this manuscript addresses an important health issue in a low resource setting, it needs a lot of work to clarify some ambiguities and be brought to a standard format for publication. Please see below my comments.

INTRODUCTION:

- Para 1, line 93please add the rate of pre-term labour in the United States.

Response to Reviewer #1: The rate of preterm labour in the United States has been added in the updated manuscript in line 93. It now appears as “The incidence of preterm labour in Sub-Saharan Africa is between 15-38%[3] as compared to Europe 5-11% [4]and 10-15% in the United States[5].

- Para 3, line 109: the authors have indicated that preterm deliveries that occur below 28 weeks are considered as inevitable abortion. According to the universally accepted definition of abortion, an abortion is a procedure in which they use medicine or surgery to r to end a pregnancy prior to 20 weeks' gestation or when the fetus born weighs less than 500 g. Please amend your sentence. You may write “These are considered as inevitable pregnancy loss.”

Response: Thanks for the caution; the sentence has been changed to “These are considered as inevitable pregnancy losses” in line 109 of the updated manuscript.

- Line 114: please replace “case-to-case” with “case-by-case”.

Response: “Case-to-case has been changed to “case-by-case” as suggested by the reviewer. The change appears in the line 115. It now appears in the revised version of the manuscript as “Though Caesarean delivery is commonly used for extreme preterm delivery in resource rich settings, the route of delivery is controversial and determined on a case-by-case basis according to the Cochrane systematic review [13]”.

MATERIALS AND METHODS:

- My general comment on this section is that there are several headings and subheadings that can be combined, or moved around to ensure a logical flow to the manuscript. For instance, “Study design” and “study design justification” need to be combined under one single heading. “Inclusion criteria”, “exclusion criteria” both can go under the “Participants”. My suggestion for the order of headings is as follows: Study design (containing design and its justification) - Study setting – Participants (containing inclusion and exclusion criteria, categorization of the participants) - Variables – Measures (tools) – Sample size calculation - Procedure (containing staff training, sampling procedure, management portions, data collection procedures) – Statistical analysis – Ethics. Of course, when combining headings/subheadings, repetitive sentences and information need to be removed and the information has to be presented in a logical order.

Response: Thanks for the advice. The method’s section has been reorganized as suggested by the reviewer. Headings and subheadings like Justification, inclusion and exclusion criteria have been removed from the revised manuscript as suggested. These changes are from line 125 to 391of the revised manuscript.

- Line 135: the study design is “prospective observation cohort”. Please amend.

Response: Thanks for the correction. The sentence has been amended. It now appears in the revised manuscript in line 135-136 as “It’s to this end that the study design is a prospective observation cohort to answer these specific study objectives”.

- Study design, Line 15o: please amend as “The study units included the labour ward, emergency obstetrics unit,….”. Same with line 153: “in the labour ward and emergency obstetrics unit,…”

Response: Thanks for the recommendation. The sentences have been amended accordingly in line 150-154 in the revised manuscript. It now appears as “The study units included the labour ward, emergency gynaecology unit, postnatal ward, labour ward operating theatre, HDU (High Dependency Unit) and the Neonatal Special Care Unit”. The units operate 24 hours per day. 

“In the labour ward and emergency gynaecology unit”, patients with signs of preterm labour were recruited and followed up to discharge from hospital.

- Lines 171 and 178, please replace “gust of liquor” and drainage of liquor” with “leakage of amniotic fluid”. Please amend throughout.

Response: “Gush of liquor and drainage of liquor” have been replaced with leakage of amniotic fluid in line 171 and 178 

- It is unclear whether the PROM was an inclusion or exclusion criteria. In “participants” section it is mentioned that “The criteria for identifying study participants was based on experience of any of the following conditions: a history of painful contractions of progressive intensity, lower back or abdominal pain, bloody mucus discharge, gush of liquor between 24 to 28 weeks of gestation.”. However, in the next page, line 181, it is indicated that “When premature rupture of membranes was suspected, a sterile speculum examination was done to exclude PROM or cord prolapse.” Please clarify.

Response: Thanks for the observation. The rationale for the speculum examination was to confirm PROM and exclude cord prolapse. This clarity has been added in the revised manuscript. It now appears in line 180-82 as “When premature rupture of membranes was suspected, a sterile speculum examination was done to confirm PROM and also to exclude cord prolapse”.

- Sample size calculation: The original sample size was calculated to be 320 women. Taking into account of a 20% loss to follow-up (n=64), the final sample size should be 384. However, you recruited 392 women. Please clarify.

Response: Since it was a prospective cohort, we had anticipated a high dropout rate during the COVID 19 pandemic. All the participants however had preterm delivery and were followed up to the end. It’s why we had 392 as opposed to 384. In a prior study in PROM after 28 weeks, some of the participants opted to deliver in other facilities and never returned. This prior experience influenced our recruitment and follow up of participants as the study was conducted during COVID-19.

- Quality control section: lines 297-300, please remove these sentences: “Data was backed up. The database was password protected and the participants’ records were kept under limited access in a lockable cabin. The research materials were kept under restricted access by only authorized staff for patient confidentiality and privacy.”

Response: Lines 297-300 have been deleted from the manuscript as recommended by the reviewer.

- Ethical consideration section: lines 305-to the end of paragraph: Please remove “Benefits and risks involved in the study were communicated to the participants. ……. not identified by names in the final report.”

Response: The content has been deleted from the manuscript as recommended by the reviewer.

RESULTS:

- Throughout the results section, the authors have reported the significance of some variables that are not actually statistically significant according to the p-value shown in the respective tables. For example, “…duration of drainage of liquor (p- value=0.05)…”.Please revise this section and amend your test report.

Response: The result’s section has been revised accordingly. Duration of drainage of liquor has been deleted as its p-value was 0.053 yet the level of significance was set for <0.05. This change is shown in line 416-17.

- Also, there are many headings and sub-headings in the “Results” section, with no particular order that make it difficult to follow the concept. Also, there is no need to explain non-significant results. For example, the heading on “Effect of Antibiotic administration on perinatal mortality” is redundant and can be removed. 

Response: “Effect of Antibiotic administration has been deleted as advised by the reviewer.

I suggest to combine headings related to each other and explain only significant findings in the text. For example, you may combine “Impact of mode of delivery on perinatal mortality” and “Impact of Maternal HIV status on pregnancy outcomes”.

Response: Since HIV didn’t have any statistically significant impact on pregnancy outcomes, this portion has been deleted from the revised manuscript.

- Line 396: please put the “95% CI 64.2-73.4” in parenthesis.

Response: This aspect has been incorporated in the revised manuscript. It now appears in line 396 as “The incidence of preterm labour between 24 to 28 weeks in the study population during this period was 68.9%, “95% CI: 64.2 – 73.4)”.

- Line 317-320: this paragraph is a bit confusing. My understanding is that during the study period, from February 2020 to September 2020, 15,346 women delivered out of which 1,916 were preterm deliveries between 24 to 28 weeks. You reported that “The incidence of preterm labour between 24 to 28 weeks in the study population during this period was 68.9%, 95% CI 64.2 – 73.4”. The incidence is very high which does not make sense. Can you please explain how you estimated the incidence? The prevalence is 12.5%, but you have indicated it was 20%! Then, the last sentence of this paragraph mentions that “The overall incidence of preterm deliveries occurring between 24 to 28 weeks of gestation during the study period at Kawempe National Referral Hospital was 2.5%.”. Please clarify and rewrite this paragraph.

Response: The study was among women with preterm labour between 24 -28 weeks. The incidence of having preterm labour among women who delivered preterms between 24-28 weeks was 68.9%. Others who delivered between 24-28 weeks had clinical presentations like preeclampsia. The incidence of preterm births during the study period was 12.4% of which preterm births between 24-28 weeks constituted 20% of this burden. Of the total deliveries that occurred during the study period, the incidence of preterm births between 24-28 weeks was 2.5%.

This paragraph now appears in the revised manuscript as “From February 2020 to September 2020, 15,346 women delivered 15,401 babies at Kawempe National Referral Hospital in Kampala, Uganda. There were 1,916 preterm deliveries during the study period giving an incidence of 12.4%. Preterm births between 24 to 28 weeks constituted 20% of the total number of preterms delivered during the study period. The overall incidence of preterm births between 24 to 28 weeks of gestation at Kawempe National Referral hospital was 2.5%. The incidence of preterm labour among women who delivered between 24 to 28 weeks in the study population during this period was 68.9%, “95% CI: 64.2 – 73.4)”. Among the reasons for those who delivered between 24 to 28 weeks of gestation yet with no preterm labour were hypertensive disorders of pregnancy (like preeclampsia), Diabetes mellitus, or congenital anomalies”.

- Line 325: to show mean and standard deviation, you may write it as 26(±5.8).

Response: The sentence has been amended as suggested by the reviewer. It now appears in the revised manuscript as “The participants were normally distributed with mean age of 26(±5.8) years.

- Line 326-327 “The median gravidity was 2 with a variance of 3.2”. T report median, you need median and interquartile range (IQR), not a variance. Please amend.

Response: The sentence has been amended as suggested by the reviewer. It now appears in the revised manuscript as “The median gravidity was 2 with an interquartile range of 3.2.

- Section “Impact of mode of delivery on perinatal mortality”, line 353: Please remove “95% CI 73.4 – 81.7)” from the text. The same section, the result of multivariate analysis would be easier to understand and interpreted by the readers if they are presented in a table indicating the reference item for each variable.

Response: Table 2 has been added to the write up and the “95% CIs from the section on “Impact of mode of delivery on perinatal mortality”, line 353 have been removed as suggested by the reviewer.

TABLES:

There are several issues with tables that need to be fixed:

- In all tables there are non-significant p-values that have been marked by an astrix (*) as significant. On the other hand, some of the significant p-values have not been marked! Please remove the * from the non-significant p-values: table 1a, 0.14*, table 1b, 0.13*, 0.053*.

Response: The astrix (*) has been removed from all non-significant p-values in the tables as suggested by the reviewer.

- Table 1a: variable “number of previous scare”, does this refer to scar on cervix or uterus? Please clarify. Also variable “Outcome of previous pregnancy”, please write all abbreviated terms in full (FSB, MSB, ENND). You may add the full words as footnote to the table.

Response: Clarity has been added to the “previous scars”. It now appears as “previous uterine scars”. All terms “FSB, MSB, ENND” have been written in full as Fresh stillbirth, Macerated stillbirth and Early Neonatal death” accordingly in the revised manuscript.

- Table 1b: variables “Labour medically induced”, “Fetal Heart Present”, “Obstetric US was done” and “Drugs given to delay labor?”, please calculate percentage for each number and add them to the table. Also, please write the ‘ultrasound (US)’ in full. Remove question mark from “Drugs given to delay labor?”

Response: The percentages have been calculated and added into the Table 1b. Ultrasound scan has been written in full. The question mark has been deleted as advised by the reviewer in the revised manuscript.

- Table 2: please write the followings in full: SVD, UTI. Also, please indicate whether the “History of vaginal bleeding” was during pregnancy or in labour.

Response: Clarity has been added into Table 2 to indicate presence of vaginal bleeding prior to onset of labour. Both SVD and UTI have been written in full as Spontaneous Vaginal delivery and Urinary tract infection respectively. 

- Table 4: please add the reference group to each variable.

Response: The reference groups have been added into the Table as recommended by the reviewer.

6. PLOS authors have the option to publish the peer review history of their article (what does this mean?). If published, this will include your full peer review and any attached files.

Do you want your identity to be public for this peer review? For information about this choice, including consent withdrawal, please see our Privacy Policy.

Reviewer #1: Yes: A/Prof Marjan Khajehei

While revising your submission, please upload your figure files to the Preflight Analysis and Conversion Engine (PACE) digital diagnostic tool, https://pacev2.apexcovantage.com/. PACE helps ensure that figures meet PLOS requirements. To use PACE, you must first register as a user. Registration is free. Then, login and navigate to the UPLOAD tab, where you will find detailed instructions on how to use the tool. If you encounter any issues or have any questions when using PACE, please email PLOS at figures@plos.org. Please note that Supporting Information files do not need this step

AUTHORS’ RESPONSE TO REVIEWS

Date: 12th June 2021

Editor’s comments

Thank you for submitting your manuscript entitled "Incidence, Associated Risk factors, and The Ideal Mode of delivery following Preterm labour between 24 to 28 weeks of gestation in a Low resource setting." to PLOS ONE. Your manuscript files have been checked in-house but before we can proceed we need you to address the following issues:

I.) Thank you for updating your data availability statement. You note that your data are available within the Supporting Information files, but no such files have been included with your submission. At this time we ask that you please upload your minimal data set as a Supporting Information file, or to a public repository such as Figshare or Dryad.

Please also ensure that when you upload your file you include separate captions for your supplementary files at the end of your manuscript.

As soon as you confirm the location of the data underlying your findings, we will be able to proceed with the review of your submission.

Response: I have attached supporting information and also added the supplementary files at the end of the manuscript.

II.) Please ensure that you refer to Table 3 in your text as, if accepted, production will need this reference to link the reader to the Table.

Response: Reference to Table 3 has been added in the revised manuscript in line 357.

18th June 2021

Editor’s comments

Thank you for submitting your manuscript entitled "Incidence, Associated Risk factors, and The Ideal Mode of delivery following Preterm labour between 24 to 28 weeks of gestation in a Low resource setting." to PLOS ONE. Your manuscript files have been checked in-house but before we can proceed we need you to address the following issues:

I.) Please kindly amend the blue text in your supporting information file to black

Response: I have amended the blue text in the supporting information. It now appears in black in the revised manuscript.

II.) Please remove the tracked changes /comments from Supporting Information.

Response: I have removed the tracked changes/ comments from the supporting information.

---

## [Decision Letter · Decision Letter 1]

5 Jul 2021

Incidence, Associated Risk factors, and The Ideal Mode of delivery following Preterm labour between 24 to 28 weeks of gestation in a Low resource setting.

PONE-D-21-00658R1

Dear Dr. Kayiga,

We’re pleased to inform you that your manuscript has been judged scientifically suitable for publication and will be formally accepted for publication once it meets all outstanding technical requirements.

Kind regards,

Antonio Simone Laganà, M.D., Ph.D.

Academic Editor

PLOS ONE

Additional Editor Comments (optional):

I carefully evaluated the revised version of this manuscript.

Authors have performed the required changes, improving significantly the quality of the paper.

Reviewers' comments:

Reviewer's Responses to Questions

**Comments to the Author**

1. If the authors have adequately addressed your comments raised in a previous round of review and you feel that this manuscript is now acceptable for publication, you may indicate that here to bypass the “Comments to the Author” section, enter your conflict of interest statement in the “Confidential to Editor” section, and submit your "Accept" recommendation.

Reviewer #1: All comments have been addressed

2. Is the manuscript technically sound, and do the data support the conclusions?

Reviewer #1: Yes

3. Has the statistical analysis been performed appropriately and rigorously? 

Reviewer #1: Yes

4. Have the authors made all data underlying the findings in their manuscript fully available?

Reviewer #1: Yes

5. Is the manuscript presented in an intelligible fashion and written in standard English?

Reviewer #1: Yes

6. Review Comments to the Author

Reviewer #1: Thank you for the revised manuscript. I trust that results of your research work will be valuable to those who are interested in the management of preterm births in low resource settings.

7. PLOS authors have the option to publish the peer review history of their article (what does this mean?). If published, this will include your full peer review and any attached files.

Reviewer #1: **Yes: **A/Prof Marjan Khajehei

---

## [Editor Report · Acceptance letter]

13 Jul 2021

PONE-D-21-00658R1 

Incidence, Associated Risk factors, and The Ideal Mode of delivery following Preterm labour between 24 to 28 weeks of gestation in a Low resource setting. 

Dear Dr. Kayiga:

I'm pleased to inform you that your manuscript has been deemed suitable for publication in PLOS ONE. Congratulations! Your manuscript is now with our production department. 

Kind regards, 

on behalf of

Dr. Antonio Simone Laganà 

Academic Editor

PLOS ONE